# Reduction in Drying Shrinkage and Efflorescence of Recycled Brick and Concrete Fine Powder–Slag-Based Geopolymer

**Xiaoming Liu [1,\*], Erping Liu [1] and Yongtong Fu [2]**

[1]  School of Civil Engineering, Central South University, Changsha 410075, China
[2]  Shenzhen Municipal Design and Research Institute Co., Ltd., Shenzhen 518029, China
\*  Correspondence: 207076@csu.edu.cn

**Abstract:** It is an effective method to prepare geopolymer with recycled brick and concrete fine powder (RP) and slag as main materials for the resource utilization of construction waste. However, its hydration products have large drying shrinkage and high efflorescence risk under normal curing conditions. Until now, the durability of recycled brick and concrete fine powder–slag-based geopolymer (RPSG) has not been well documented, such as drying shrinkage and efflorescence. In this study, the effects of slag content, alkali equivalent and modulus on the durability properties of RPSG were evaluated. The results show: (1) Slag can significantly reduce the drying shrinkage and efflorescence of RPSG. (2) The potential for the efflorescence of RPSG increases with increasing alkali equivalent. The drying shrinkage of RPSG increases with the increase of alkali equivalent in the case of a low alkali equivalent (6 wt.% in this paper) and decreases with the increase of alkali equivalent in the case of a high alkali equivalent. (3) The drying shrinkage of RPSG increases with increasing modulus. In contrast, the degree of efflorescence decreases with increasing modulus. In this study, RP-S45-M1.3N6 (slag content: 45 wt.%; alkali equivalent: 6 wt.%; modulus: 1.3) is the best proportional design for RPSG with excellent durability. Compared to RP-S0-M1.3N6, the drying shrinkage of RP-S45-M1.3N6 is reduced by 76.32%, the capillary porosity is reduced by 60.9%, the visual efflorescence is significantly alleviated, and the early pH value is reduced by approximately 2.0. This paper systematically analyzed the drying shrinkage pattern and the efflorescence pattern of RPSG, which has a positive significance for promoting the recycling of RP from construction waste.

**Keywords:** recycled brick and concrete powder; slag; geopolymer; drying shrinkage; efflorescence





## 1. Introduction

With the rapid development of global urbanization, the emission of construction waste is increasing rapidly. According to the statistics, current construction waste accounts for about 30% of urban solid waste [1–5], and more than 10 billion tons of construction waste are generated worldwide every year, 80% of which are waste bricks and concrete [6,7]. At present, the treatment method for construction waste is mainly stacking and landfill, which not only wastes land but also seriously endangers the natural environment [8]. On the other hand, the exploitation of natural building materials in China is protected strictly. Therefore, the recycling of construction waste can not only protect the environment but also save resources and alleviate the current shortage of natural building materials.

In recent years, recycled aggregates produced from construction waste have been widely used in recycled concrete [9–11], recycled mortar [12,13], roadbase [14] and other fields. In the process of producing recycled aggregate from construction waste, a large amount of fine powder with a size of less than 0.16 mm, which is called recycled fine powder, is produced. Compared with untreated construction waste, the long-term accumulation of recycled fine powder will not only occupy land but also produce dust, pollute water resources and do greater harm to the environment [15].

Geopolymers have many advantages, such as faster hardening, higher strength, excellent acid and alkali resistance, lower carbon emission, and are considered to be the most potential low-carbon green cementitious material to replace ordinary Portland cement (OPC). The reaction mechanism is that the silicon oxygen bond and aluminum oxygen bond in the geopolymer precursor materials break in an alkaline environment to form $AlO_4$ and $SiO_4$ tetrahedral monomers, and then polymerize and recombine to form a three-dimensional network structure [16,17]. The preparation of geopolymers from some wastes, such as metakaolin, slag, fly ash, rice husk ash, etc., with aluminosilicate as the main component is a research frontier [16,17]. As a material mainly composed of silica and alumina, recycled fine powder has the potential to develop geopolymers. At present, the use of recycled fine powder as a precursor material of geopolymers has been studied [18–23]. Meanwhile, the mechanical properties of recycled fine powder-based geopolymers have been widely qualified and have great potential to replace OPC [22–25].

Drying shrinkage is one of the main reasons that durability and applicability of concrete structures are reduced [26]. The shrinkage of concrete will cause tensile stress due to the restraint of itself and external conditions. When the shrinkage is large, it will cause harmful crack formation and eventually lead to serious crack damage of concrete [27]. Compared with OPC, the geopolymer will produce greater dry shrinking under dry conditions, restricting its actual engineering applications [28–30]. Previous studies [31–33] have shown that the shrinkage of geopolymer concrete is 2–4 times that of ordinary concrete. Many scholars have studied how to reduce the drying shrinkage of alkali-activated binders in terms of alkali equivalents, activator types, mineral admixtures, chemical additives and curing conditions [31]. Duxson et al. [34,35] found that Si/Al and gel structure are two main parameters of sodium-based geopolymers, which have a direct impact on shrinkage. Deb et al. [36] showed that both incorporating a small amount of fly ash and reducing the modulus of modified sodium silicate reduced the drying shrinkage of geopolymer concrete cured at room temperature. Ma et al. [37] also found that lowering the modulus of activators can effectively reduce the drying shrinkage of geopolymer concrete. Additives are one of the effective means to reduce the drying shrinkage of geopolymer concrete. There were popular additives, such as magnesium oxide [38], calcium oxide [39], gypsum [40], expansion agent [41] and absorbent polymer materials [42]. Hardjito et al. [43] found that increasing the curing temperature is another measure to reduce the drying shrinkage of geopolymers. Palacios et al. [44] found that the drying shrinkage of geopolymer can also be reduced by increasing the curing humidity.

However, the chemical nature of recycled fine powders differs significantly from that of other types of precursor materials, and it remains unknown whether these strategies will be effective in reducing the drying shrinkage of recycled fine powder-based geopolymers.

In geopolymer systems, efflorescence is mainly caused by alkali metal cations freeing to the surface of the material and reacting with carbon dioxide to form substances such as sodium carbonate and sodium bicarbonate [45,46]. The highly alkaline reaction conditions of geopolymers will inevitably result in a much higher degree of efflorescence than that of OPC. Efflorescence not only affects the esthetics of the building material, but also reduces the strength of the material and affects the durability of the structure. The efflorescence of alkali-activated binders severely limits their practical engineering applications [47–49]. Therefore, it is crucial to limit the efflorescence of geopolymers. Zhang et al. [50] found that the addition of 20% slag to fly ash-based cementitious materials were able to reduce the rate of efflorescence in specimens, while efflorescence could also be inhibited by high-temperature curing. A similar result was found by Kani et al. [51], who showed that the addition of alumina-rich precursors or the addition of slag was effective in inhibiting alkali efflorescence. In contrast, Xiao et al. [52] found that the addition of less than 50% mass of slag to fly ash did not significantly change the degree of efflorescence in the specimens, which showed higher levels of efflorescence with slag content above 50%. Therefore, the relationship between the amount of slag and efflorescence is not entirely positive. Zhang et al. [50] found that soluble silica reduced the porosity of the specimens to inhibit

efflorescence at the same alkali equivalent, and also pointed out that the efflorescence degree of NaOH-activated geopolymers was lower than that of sodium silicate-activated geopolymers. With regard to the effect of alkali equivalent on the efflorescence properties, Saha et al. [53] concluded that reducing the alkali equivalent would reduce the degree of efflorescence, but when the alkali equivalent was less, it would again affect the strength of the specimen, so a relative compromise was needed. Wu et al. [54] suggested that water evaporation is an important factor affecting the efflorescence of fly ash-based geopolymers. Tan et al. [55] first investigated efflorescence mitigation in recycled fine powder-based geopolymers and demonstrated that both slag and metakaolin could mitigate the degree of efflorescence in recycled fine powder based-geopolymers.

As can be seen from the above, recycled fine powder-based geopolymers have great potential to replace OPC. However, little research has been reported on the durability of recycled fine powder-based geopolymers in terms of drying shrinkage and efflorescence, which restricts the promotion and application of recycled fine powder-based geopolymers.

In this paper, recycled brick and concrete fine powder–slag-based geopolymer (RPSG) was prepared by using recycled brick and concrete fine powder (RP) and slag. The effects of different slag contents, alkali equivalents and moduli on drying shrinkage and efflorescence were studied. X-ray diffraction (XRD), Fourier transform infrared spectroscopy (FTIR), thermogravimetry (TG) and scanning electron microscope (SEM) were used to analyze the hydration products and their microstructure. The reaction mechanism of RP and slag was discussed, and the drying shrinkage and efflorescence patterns of the RPSG were systematically analyzed.

## 2. Research Significance

Firstly, this paper systematically investigates the drying shrinkage pattern and the efflorescence of RPSG, which is a useful addition to the research related to the durability performance of recycled fine powder-based geopolymers. The research results of this paper are beneficial to promote the practical application of recycled fine powder-based geopolymers. As a low-carbon green cementitious material, RPSG could replace OPC in part of the engineering field, which has great practical significance to the sustainable development of cement industry.

## 3. Materials and Methods

### 3.1. Materials

Recycled brick and concrete fine powder (RP): it was obtained from recycled brick and concrete fine aggregate (0–4.75 mm), mainly including clay bricks, mortar, tiles, ceramics, etc. It was dried to constant weight at 105 °C, and then passed through a standard sieve with 2.36 mm. Then, it was crushed and sieved by a small laboratory crusher to obtain fine powder with a particle size less than 0.16 mm. The process is shown in Figure 1. Considering the crushing efficiency, the crushing time was 40 s, and the powder yield was 56.41%. The main chemical composition of RP can be seen in Table 1. $SiO_2$ and $Al_2O_3$ are the two main components. The XRD results, morphology and particle size distribution are shown in Figures 2–4, respectively.

Slag: S95 grade granulated blast furnace slag with a density of 2.9 $kg/cm^3$, specific surface area of 412 $m^2/kg$ and 28-day activity index of 95.5% was used. The main chemical components can be seen in Table 1.

Sodium silicate solution: The liquid sodium silicate solution with a modulus of 3.2 $kg/cm^3$ and water content of 65% was used in this test.

Sodium hydroxide: NaOH used in the test was flake NaOH with a purity of 99.0%.

Water (W): Except for the pH value test, which used purified water, the water used for the tests was laboratory tap water.

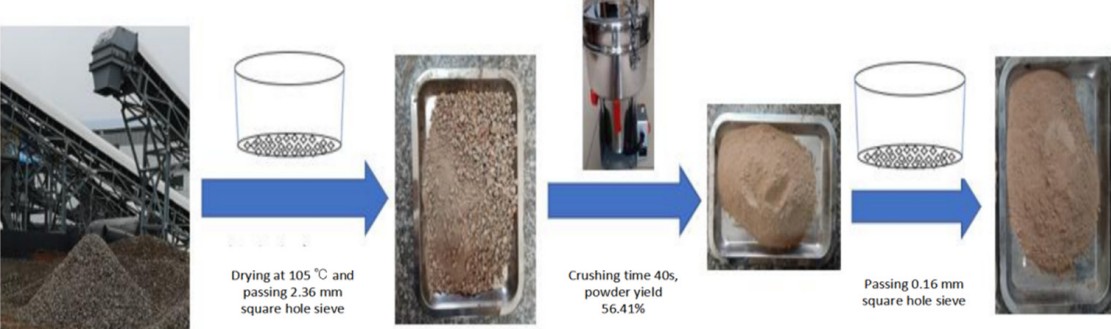

**Figure 1.** The preparation process of RP.

**Table 1.** The chemical compositions of RP and slag.

|  | SiO$_2$ | Al$_2$O$_3$ | CaO | Fe$_2$O$_3$ | K$_2$O | MgO | Na$_2$O | TiO$_2$ | Others |
|---|---|---|---|---|---|---|---|---|---|
| RP | 49.4 | 20.2 | 17.29 | 4.7 | 2.1 | 1.4 | 1.3 | 0.6 | 3.0 |
| S | 30.3 | 14.2 | 39.3 | 0.7 | 0.4 | 7.1 | 0.4 | 0.5 | 7.1 |

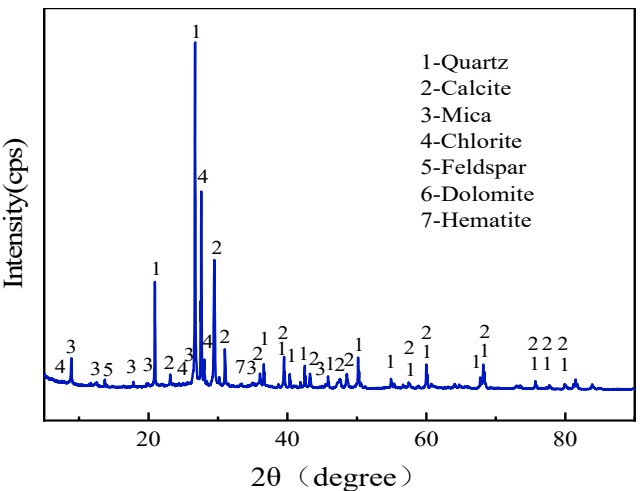

**Figure 2.** The XRD patterns of RP.

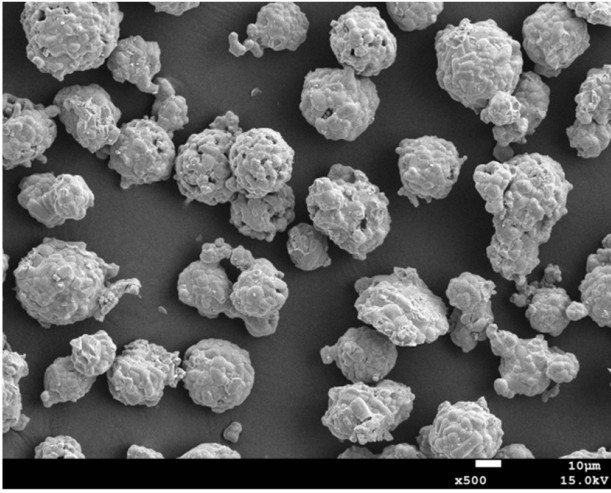

**Figure 3.** The SEM images of RP [56].

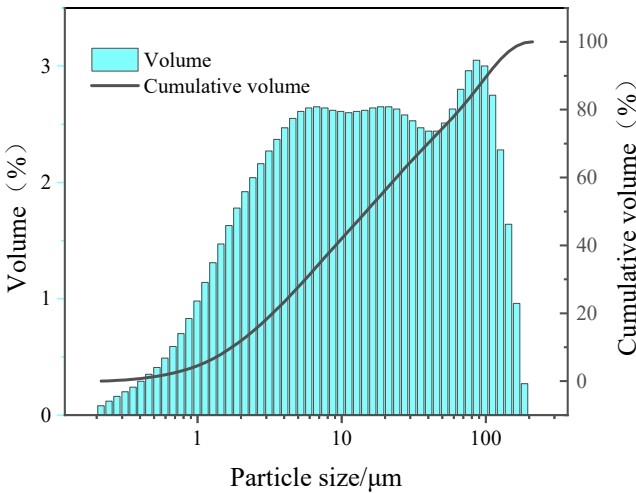

**Figure 4.** The particle size distribution of RP.

### 3.2. Mixture Proportion Design

In order to study the influence of slag content on the properties of RPSG, the specimens were prepared with slag contents of 0 wt.%, 15 wt.%, 30 wt.% and 45 wt.% (calculated by the total mass of RP and slag), and the fixed modulus and alkali equivalent were 6 wt.% and 1.3, respectively. In order to study the effect of modulus and alkali equivalent on the performance of RPSG, the fixed slag content was 30 wt.%, and the moduli were adjusted to 0.9, 1.1, 1.3, 1.5 and 1.7, and the alkali equivalents were 3 wt.%, 6 wt.%, 9 wt.%, 12 wt.% and 15 wt.%, respectively. The mixture proportions are shown in Table 2. The specimens were numbered according to the slag content, modulus and alkali equivalent. For example, for RP-S0-M1.3N6, RP represents recycled brick and concrete fine powder, S0 represents 0 wt.% slag content, M1.3 represents 1.3 modulus of alkali activator solution, and N6 represents 6 wt.% alkali equivalent of composite cementitious system. The water binder ratio in this test was all 0.35. In order to investigate the hydration products and microstructure of RPSG, specimens RP-S0-M1.3N6 and RP-S30-M1.3N6 were selected for further micro-analysis.

**Table 2.** The mixture proportions of RPSG.

| Mix ID | $^a$ W/$^b$ B | RP | Slag | $^c$ W | NaOH | $Na_2SiO_3$ |
|---|---|---|---|---|---|---|
| RP-S0-M1.3N6 | 0.35 | 500.00 | 0.00 | 107.95 | 23.09 | 142.42 |
| RP-S15-M1.3N6 | 0.35 | 425.00 | 75.00 | 107.95 | 23.09 | 142.42 |
| RP-S30-M1.3N6 | 0.35 | 350.00 | 150.00 | 107.95 | 23.09 | 142.42 |
| RP-S45-M1.3N6 | 0.35 | 275.00 | 225.00 | 107.95 | 23.09 | 142.42 |
| RP-S30-M0.9N6 | 0.35 | 350.00 | 150.00 | 132.75 | 27.90 | 98.60 |
| RP-S30-M1.1N6 | 0.35 | 350.00 | 150.00 | 120.35 | 25.49 | 120.51 |
| RP-S30-M1.5N6 | 0.35 | 350.00 | 150.00 | 95.55 | 20.69 | 164.33 |
| RP-S30-M1.7N6 | 0.35 | 350.00 | 150.00 | 83.15 | 18.28 | 186.24 |
| RP-S30-M1.3N3 | 0.35 | 350.00 | 150.00 | 141.48 | 11.54 | 71.21 |
| RP-S30-M1.3N9 | 0.35 | 350.00 | 150.00 | 74.43 | 34.63 | 213.63 |
| RP-S30-M1.3N12 | 0.35 | 350.00 | 150.00 | 40.91 | 46.18 | 284.84 |
| RP-S30-M1.3N15 | 0.35 | 350.00 | 150.00 | 7.38 | 57.72 | 356.06 |

$^a$ W (Water content) = water content of $Na_2SiO_3$ solution + $^c$ W. $^b$ B = RP + S + solid content of $Na_2SiO_3$ solution. $^c$ W—additional water.

### 3.3. Test Method

(1) Dry shrinkage test: The dry shrinkage test was conducted in accordance with the Test Method for Dry Shrinkage of Cement Mortar in China (JC/T 603-2004), and the mix proportions are shown in Table 2. The modified sodium silicate solution with the appropriate modulus and alkali equivalent was prepared 24 h in advance, then the

modified sodium silicate solution was mixed with RP and slag and poured into the cement mortar mixer. After stirring on low speed for 3 min, a scraper was used to scrape up the slurry from the bottom of the mortar mixing pot and stir manually for 2 min, then stirring continued on high speed for 3 min. The mixed slurry was rapidly poured into a 25 mm × 25 mm × 280 mm mold with embedded metal nails at both ends. After 20 s of vibrating, it was cured with plastic film at room temperature for 48 h, then stripped and transferred to 20 ± 2 °C and 50 ± 5% relative humidity for 28 days of dry shrinkage curing. BC-300 cement length meter was used to measure the size change, and the length was measured once every 24 h. Each group contained 3 specimens, and the average value of the results was analyzed.

(2) Capillary porosity test: According to Table 2, the rectangular specimens with 40 mm × 40 mm × 40 mm were wrapped by film under standard conditions for 28 days, and then crushed into pieces. The capillary porosity of the pieces was determined using the saturated water method [45].

(3) Rapid efflorescence test: The specimen preparation method was the same as that of the capillary porosity test. The specimens were wrapped in plastic film and maintained for 28 days under standard conditions, then placed in plastic containers with an appropriate amount of tap water so that the bottom of the specimen was immersed in water for approximately 5 mm. A camera was used to record the images of the specimens over time. The temperature and relative humidity during the experiment were 25 °C and 55%, respectively. During the test, the proper amount of water was added to the plastic container every 5 h to maintain a fixed water level at the bottom of the specimen. After 7 days, the addition of water to the container was stopped, and the water in the container underwent complete evaporation. The white crystals and broken epidermis on the surface of the specimens were scraped out with a spatula and then weighed.

(4) PH value test: The specimens were crushed into granules from 28 days of film curing. Solid particles with a size range of 0.6–1.18 mm were selected, and then immersed in purified water. The pH value of the immersion solution was measured every 10 min for the first hour. After 1 h, measurements were taken every 1 h until the pH value of the immersion solution did not modify significantly and then stopped. In this experiment, the ratio of solid to liquid was 1:50.

(5) XRF: 3 g powder specimens were prepared. An X-ray fluorescence spectrometer was used, and the test mode was in the form of oxide.

(6) XRD: The specimen at the specified age was first crushed into small pieces. The pieces < 1.18 mm were taken and immersed in anhydrous alcohol for 7 days to terminate the hydration process, then dried at 50 °C to constant weight. Finally, the specimens were ground into powder, and 1 g was selected for the XRD test after passing 200 mesh sieve. X-ray diffractometer with scanning speed of 2°/min and scanning angle range of 5–90° was adopted.

(7) FTIR: The specimen preparation method was the same as that of XRD. The test mode was conventional powder tablet. The wavenumber range was 400–4000 cm$^{-1}$.

(8) TG: The specimen preparation method was the same as that of XRD. 20 mg powder specimen was selected and tested with a thermogravimetric analyzer. The test temperature range was room temperature ~1000 °C, the test gas atmosphere was nitrogen, and the heating rate was 10 °C/min.

(9) SEM: Block specimens with thickness less than 1 cm, diameter ≤ 1 cm and relatively flat fracture surface were selected and dried at 50 °C to constant weight. A scanning electron microscope was used to test the morphology.

## 4. Results and Discussion

### 4.1. Drying Shrinkage

4.1.1. Effect of Slag Content on Drying Shrinkage

Figure 5 shows the drying shrinkage of RPSG with different slag contents within 28 days. It can be seen that the drying shrinkage decreases with the increase of slag content.

The 28-day drying shrinkage of RP-S0-M1.3N6, RP-S15-M1.3N6, RP-S30-M1.3N6 and RP-S45-M1.3N6 are 2.98%, 2.57%, 0.93% and 0.70%, respectively. The drying shrinkage of RPSG with slag content of 15 wt.%, 30 wt.% and 45 wt.% are 13.53%, 68.69% and 76.32% lower than that of RPSG without slag. The drying shrinkage value of RPSG decreases with the increase of Ca/Si, which is contrary to the traditional conclusion that the drying shrinkage value of alkali-activated cementitious materials increases with the increase of Ca/Si [57], such as alkali-activated fly ash/slag, Kaolin/slag, silica fume/slag and other cementitious systems. It is noted that the RPSG with 60 wt.% slag content had a large drying shrinkage value, and cracking occurred on the third day after forming.

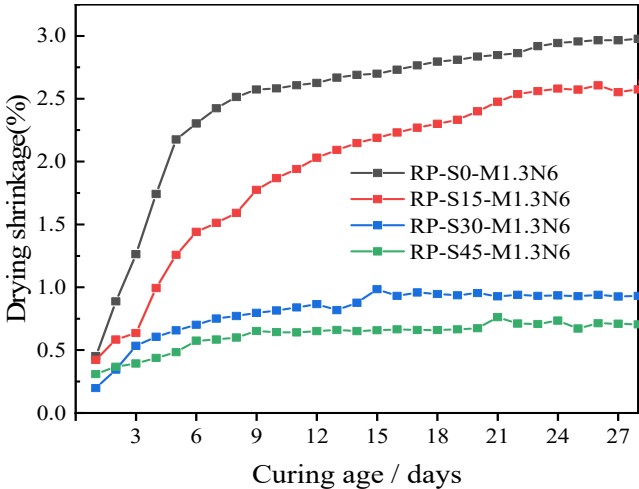

**Figure 5.** Drying shrinkage results of RPSG with different slag contents.

The capillary pressure theory shows that the essence of shrinkage is that the free water evaporates from the saturated mesoporous pores in a dry environment, and the surface tension in the pore solution forms capillary pressure. Thus, the curved moon surface of liquid–gas was formed, resulting in the reduction of slurry volume. This mechanism is applicable to alkali-activated cementitious materials [58]. The pore size distribution is the main factor affecting its drying shrinkage. However, further analysis is required for the cementitious system with two or more precursors, and a variety of factors should be considered [59]. According to the Mackenzie Bentz equation, the parameters affecting the drying shrinkage of RPSG mainly include bulk modulus, bulk modulus of solid material, pore saturation and capillary tension of liquid phase [60]. Capillary tension is mainly composed of surface tension, contact angle and Kelvin radius. The surface tension and contact angle of the pore solution are the same for different mixtures under the same humidity and are mainly affected by the pore diameter. According to the above theory, the effective factors of drying shrinkage caused by capillary tension are pore diameter, pore saturation and bulk modulus [2].

Slag can promote the production of hydrophilic products such as NASH and CASH, making the slurry denser, so as to improve the bulk modulus of the slurry and the ability of the slurry to resist drying shrinkage and deformation. Slag can refine pore structure, reduce pore size and increase capillary tension. The particle size of slag particles is smaller than that of RP. Slag particles are dispersed to the surface of RP particles during mixing. After hydration, the products can fill pores and form closed pores, improving pore saturation [59]. It can be seen from the above analysis that the drying shrinkage result is the compound action of the bulk modulus, pore diameter and pore saturation. The results show that bulk modulus is the dominant factor affecting the drying shrinkage of RPSG when the slag content is small. Therefore, slag can reduce the drying shrinkage of RPSG. When the slag content is high, the dominant factors affecting the drying shrinkage are pore diameter and pore saturation [61].

The activity of RP under alkaline conditions is much lower than that of slag. After slag replaces RP, the consumed water through chemical reaction and the active $SiO_2$ in sodium silicate solution increase, and the residual water and active $SiO_2$ in the slurry decrease. Relevant studies [30] show that silica gel with high water content is easy to dehydrate under dry conditions, resulting in larger shrinkage, while the addition of slag can reduce the water content in the slurry and consume some sodium silicate, thus reducing the drying shrinkage. In addition, RP can be used as an "internal curing agent" to reduce the drying shrinkage of alkali-activated slag cementitious systems due to its internal porous and water absorption [2,62]. Therefore, slag can significantly reduce the drying shrinkage of RPSG.

4.1.2. Effect of Alkali Equivalent on Drying Shrinkage

Figure 6 shows the drying shrinkage of RPSG with different alkali equivalents within 28 days. The test results show that the drying shrinkage does not have a certain positive correlation with the alkali equivalent. The 28-day drying shrinkage of RP-S30-M1.3N3, RP-S30-M1.3N6, RP-S30-M1.3N9, RP-S30-M1.3N12 and RP-S30-M1.3N15 are 0.40%, 0.93%, 0.83%, 0.57% and 0.18%, respectively. When the alkali equivalent is small (In this paper, the alkali equivalent is less than 6 wt.%), drying shrinkage increases with the increase of alkali equivalent. This is consistent with some current studies on alkali-activated binders. It can be explained that the increase of alkali concentration increases the proportion of pores and gel pores in the slurry, reduces the content of large pores [62] and increases the capillary tensile stress. In addition, a certain alkaline environment is the basis of slag hydration, and an appropriate increase in alkali equivalent can promote slag hydration [63]. Therefore, for the low alkali equivalent specimens, slag hydration is incomplete, and part of the slag that is not hydrated fills the large-sized pores of the recycled brick mix microfine, which slows down water evaporation. At the same time, slag that is not hydrated has a "microskeleton" effect, which can also reduce the drying shrinkage performance of the specimen. These reasons together contribute to the low alkali equivalent specimen drying shrinkage being small. When the alkali equivalent is large, drying shrinkage decreases with the increase of alkali equivalent. The specimens with high alkali equivalent are unstable, as more alkali will promote the precipitation of $Ca(OH)_2$ and the formation of low Ca/Si gel, reduce mesoporous proportion of the matrix and reduce drying shrinkage of the specimen. For example, the drying shrinkage of RP-S30-M1.3N9, RP-S30-M1.3N12 and RP-S30-M1.3N15 decreased by 11.33%, 39.27% and 80.71%, respectively, compared with RP-S30-M1.3N12. In addition, the porosity of the RPSG system is higher than that of the traditional alkali-activated cementitious system, and there are more macropores in the slurry. Although it has higher activity under high alkali equivalent, it cannot actually consume the excessive alkali component in the system, and the excessive alkali will generate white crystals in the macropores in the slurry and on the surface of the specimen under a dry environment. After crystallization is generated in the internal pores of the slurry, a certain amount of crystallization pressure will be generated on the pore wall to offset the capillary tensile stress caused by the mesoporous pores so that the drying shrinkage value of the specimen is reduced and even expanded with the extension of the curing age. It can be well verified from the drying shrinkage results of the specimen RP-S30-M1.3N15. It can be seen from Figure 6 that the specimen RP-S30-M1.3N15 with 15 wt.% alkali equivalent has a large shrinkage in the early stage, but the shrinkage value decreases with the increase of age. It can be observed that white alkali crystals are separated from the slurry of RPSG, and some visible cracks appear on the surface of the specimen. These cracks are not shrinkage cracks caused by capillary tension but expansion cracks caused by pressure formed by free alkali crystallization. However, specimens with high alkali equivalents are very unstable and prone to large cracks under ambient conditioning conditions, contributing to a reduction in specimen strength. Therefore, increasing the alkali equivalent is not a good measure to reduce the drying shrinkage of RPSG.

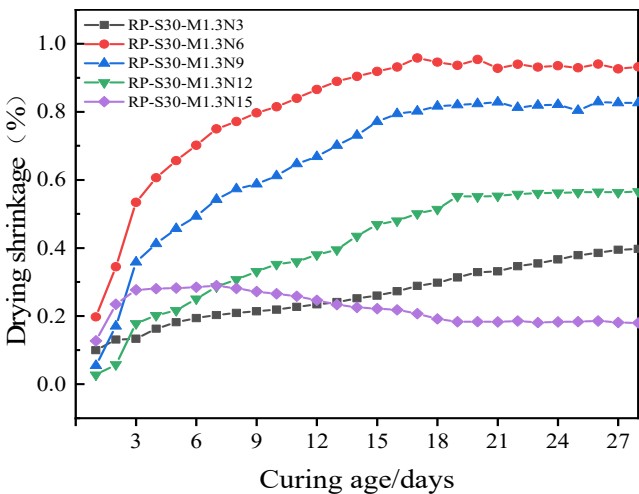

**Figure 6.** Drying shrinkage results of RPSG with different alkali equivalents.

### 4.1.3. Effect of Modulus on Drying Shrinkage

Figure 7 shows the drying shrinkage of RPSG with different moduli within 28 days. The test results show that the drying shrinkage has a certain positive correlation with the modulus. When the alkali equivalent is constant, drying shrinkage increases with the increase of modulus, which is consistent with the conclusions drawn by other studies [29]. The 28-day drying shrinkage of RP-S30-M0.9N6, RP-S30-M1.1N6, RP-S30-M1.3N6, RP-S30-M1.5N6 and RP-S30-M1.7N6 are 0.42%, 0.82%, 0.93%, 1.17% and 1.32%, respectively. For the RPSG with fixed alkali equivalent, the main factor affecting drying shrinkage is the pore distribution in the binders. When the alkali equivalent is constant, the content of sodium silicate increases relatively with the increase in modulus. Generally speaking, the increase of sodium silicate will lead to the decrease of the total porosity of the slurry and the increase of mesopore ratio, while the increase of mesopore volume will lead to the increase of capillary stress, thus increasing the drying shrinkage of the specimen. In addition, excessive sodium silicate will not react completely, and it is very easy to lose water under dry conditions, resulting in dry shrinkage of the slurry.

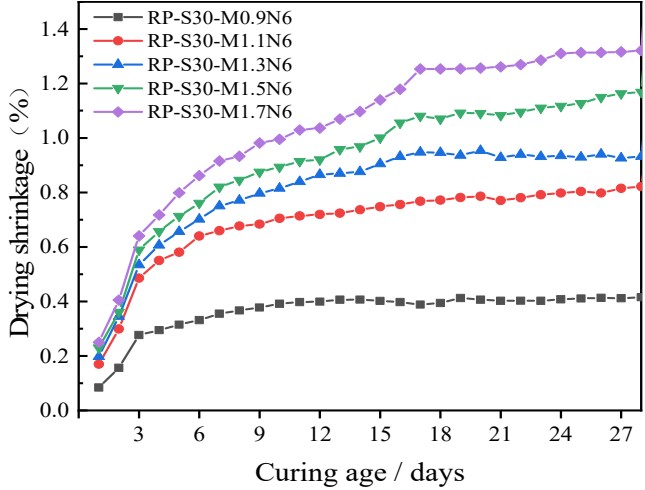

**Figure 7.** Drying shrinkage results of RPSG with different moduli.

From the above analysis, it is clear that RPSG has large drying shrinkage, with maximum drying shrinkage reaching 2.98% after 28 days, which is far higher than the drying shrinkage of metakaolin-based polymer (approx. 0.4–0.6%) [64]. However, the drying shrinkage of RPSG could be reduced by optimizing the mix ratio, such as the specimen RP-S30-M1.3N3, whose drying shrinkage is only 0.40%

### 4.2. Capillary Porosity

#### 4.2.1. Effect of Slag Content on Capillary Porosity

The permeability of the specimen is one of the important factors affecting its efflorescence, and capillary porosity is an important indicator for evaluating the permeability of the specimen. Figure 8 shows the capillary porosity of RPSG with different slag contents. It can be seen from Figure 8 that the capillary porosity of RPSG decreases with the increase of slag content. The capillary porosity of the specimens RP-S0-M1.3N6, RP-S15-M1.3N6, RP-S30-M1.3N6, RP-S45-M1.3N6 are 0.33%, 0.24%, 0.18% and 0.13%, respectively. Compared with the specimens without slag, the capillary porosity of the specimens with 15 wt.%, 30 wt.% and 45 wt.% slag content decreased by 27.3%, 45.5% and 60.6%, respectively. This is because the incorporation of slag can introduce more active calcium components into the matrix, which significantly increases the content of active components, thereby promoting the formation of hydration products such as CSH, CASH and NASH. The generated hydration products can fill the pores of the RPSG and improve the density of the slurry, which in turn leads to a decrease in the capillary porosity of the specimen. In addition, unreacted slag can also fill the pores of the RPSG, improve the density of the slurry and reduce the capillary porosity of the slurry.

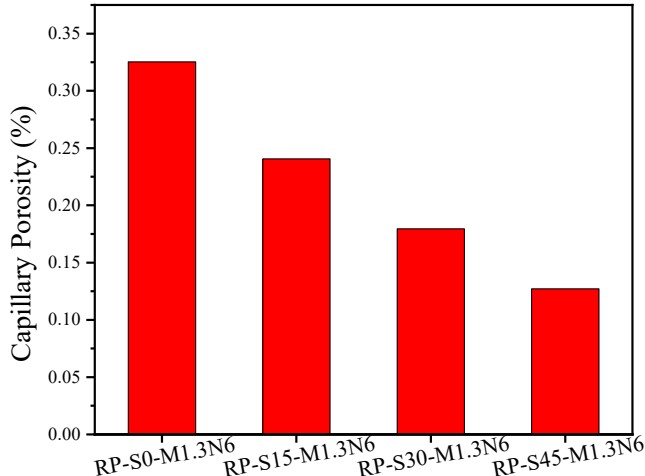

**Figure 8.** Capillary porosity of RPSG with different slag contents.

#### 4.2.2. Effect of Alkali Equivalent on Capillary Porosity

Figure 9 shows the capillary porosity of RPSG with different alkali equivalents. As can be seen from Figure 9, the capillary porosity of RPSG does not have a simple linear relationship with the alkali equivalent. The capillary porosity of specimen RP-S30-M1.3N6 was reduced by 48.90% compared to that of specimen RP-S30-M1.3N3, while the capillary porosity of specimens RP-S30-M1.3N9 and RP-S30-M1.3N12 remained essentially unchanged compared to that of specimen RP-S30-M1.3N6. Meanwhile, the capillary porosity of RP-S30-M1.3N15 increased by 50.5% compared with that of RP-S30-M1.3N12. The silica and aluminum components in RP need to be dissolved and polymerized in an alkaline environment to form NASH hydration gel. The active CaO in slag also needs an alkaline environment to dissolve and polymerize to form CSH and CASH gels. Properly increasing the alkali equivalent and improving the alkaline environment of the geopolymer reaction can promote the formation of more NASH, CSH and CASH gels, thereby improving the compactness of the specimen. When the alkali equivalent exceeds a certain threshold, the strong alkali environment will not only inhibit the dissolution of the silicon–alumina components in the RP, but also guide the active CaO in slag to form precipitation and adhere to the particle surface, hindering the progress of the RPSG hydration reaction [65]. Therefore, continuously increasing the alkali equivalent cannot continuously improve the compactness of the specimen and may even damage the compactness of the specimen.

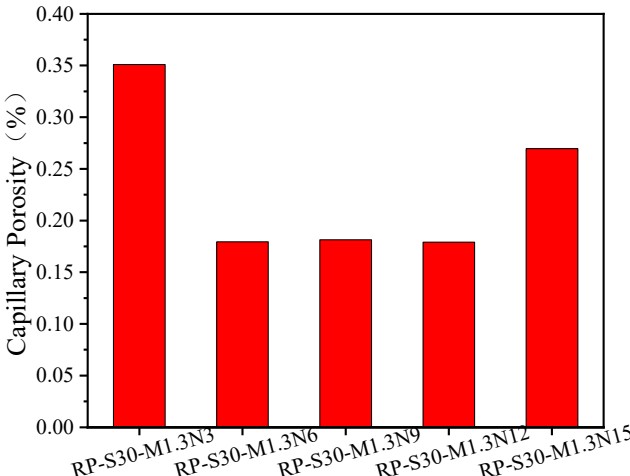

**Figure 9.** Capillary porosity of RPSG with different alkali equivalents.

### 4.2.3. Effect of Modulus on Capillary Porosity

Figure 10 shows the capillary porosity of RPSG with different moduli. It can be seen from Figure 10 that within a certain range, the capillary porosity of RPSG decreases with the increase of the modulus. Compared to the capillary porosity of specimen RP-S30-M0.9N6, the capillary porosity of specimens RP-S30-M1.1N6, RP-S30-M1.3N6, RP-S30-M1.5N6 and RP-S30-M1.7N6 was reduced by 4.9%, 10.8%, 19.4% and 19.4%, respectively. This is because the high modulus modified sodium silicate solution contains more active $SiO_2$, which is conducive to the formation of a denser gel and thus reduces the capillary porosity of the specimens [66]. Meanwhile, the hydration reaction of RPSG requires a certain alkaline environment, and the weak alkaline environment is not conducive to the formation of hydration and gelation products such as CSH, CASH and NASH. Therefore, high modulus RPSG may have higher capillary porosity.

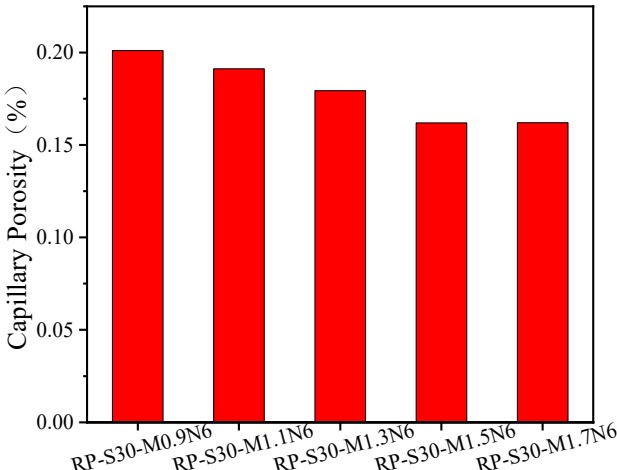

**Figure 10.** Capillary porosity of RPSG with different moduli.

### 4.3. Visual Efflorescence

### 4.3.1. Effect of Slag Content on Visual Efflorescence

Visual efflorescence can intuitively reflect the degree of efflorescence of the specimen. Figure 11 shows the visual efflorescence changes of RPSG with different slag contents at 1 day, 3 days and 7 days. It can be seen from Figure 11 that after 1 day of immersion, the white crystalline substance first appears around the top of the specimen RP-S0-M1.3N6 without slag addition, while there is no obvious change around the top of the specimens RP-S15-M1.3N6, RP-S30-M1.3N6 and RP-S45-M1.3N6. After 3 days of immersion, white

crystalline substances precipitated around and at the top of the specimen RP-S0-M1.3N6, while the specimens RP-S15-M1.3N6 and RP-S45-M1.3N6 only showed a small circle of crystalline substances at the bottom above the immersed part. However, crystallization appeared on the top of the specimen RP-S30-M1.3N6 with a slag content of 30 wt.%. This is due to the joint porosity between the top and bottom of the specimen, which causes the abnormal phenomenon that the surface of the specimen crystallizes prematurely. After 7 days of immersion, the surface of specimen RP-S0-M1.3N6 starts peeling on the surface of the crystallization pressure gauge, and part of the paste swells or falls off. This can be attributed to the fact that crystals are formed when alkaline-rich solutions inside or on the surface of the specimen pores react with $CO_2$ and continue to migrate with the moisture. The crystals continue to grow, generating a crystallization pressure. When the pressure exceeds the internal tensile strength of the specimen, cracks and rupture occurs [67]. The degree of efflorescence of other specimens is not significantly different from that of 3 days. It can be concluded from the test phenomena that the addition of slag can significantly reduce the visual efflorescence of RPSG and reduce the visual efflorescence rate.

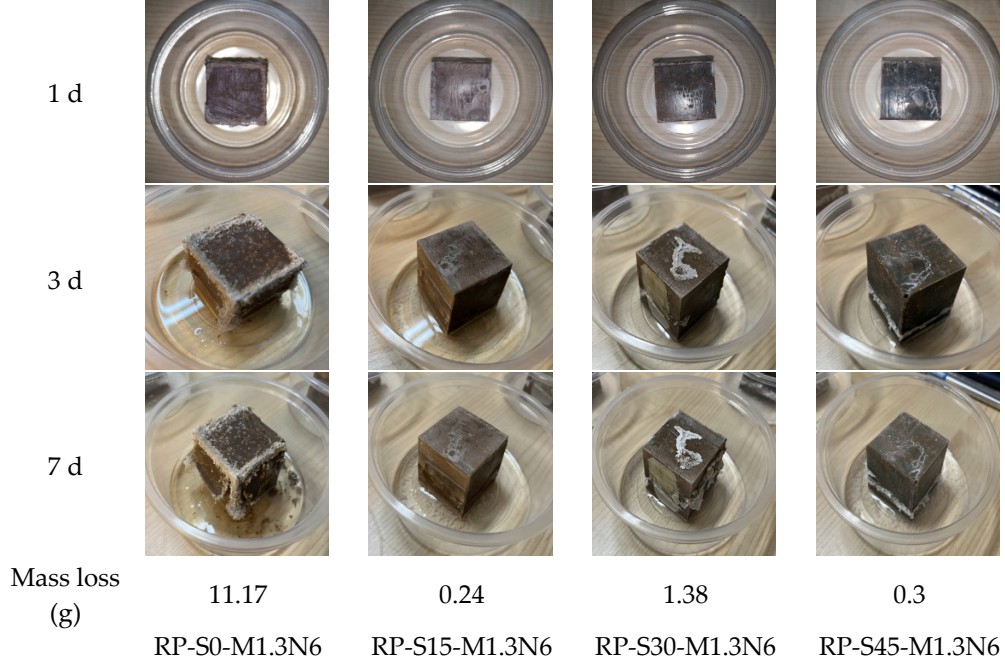

| | | | | |
|---|---|---|---|---|
| Mass loss (g) | 11.17 | 0.24 | 1.38 | 0.3 |
| | RP-S0-M1.3N6 | RP-S15-M1.3N6 | RP-S30-M1.3N6 | RP-S45-M1.3N6 |

**Figure 11.** Visual efflorescence of RPSG with different slag contents.

Visual efflorescence can intuitively show the phenomenon of efflorescence when the specimen is immersed in water at the bottom, but this test method cannot quantify the degree of efflorescence in the slurry. Figure 11 shows the mass loss results of RPSG with different slag contents due to efflorescence. The mass loss of RP-S0-M1.3N6 was the highest, reaching 11.17 g, and the results were in good agreement with visual efflorescence. Unlike the decrease in capillary porosity with increasing slag content as shown in Figure 8, specimen RP-S15-M1.3N6 showed minimal visual efflorescence and mass loss. At the same time, there is an obvious macro-crack above the submerged part at the bottom of the specimen. This is because alkali-activated materials can occur not only on the surface of the specimen but also in the larger capillary pores on the surface or cracks in the interior of the specimen. Efflorescence will produce crystallization pressure and force cracks to expand further, which will further enhance the phenomenon of efflorescence [45]. At the same time, the appearance of efflorescence substances will also be different due to the speed of evaporation, so the efflorescence substances cannot be completely scraped out during the test. In addition, the visual image of efflorescence will also have certain differences due to the different shapes of the crystalline substances.

### 4.3.2. Effect of Alkali Equivalent on Visual Efflorescence

Figure 12 shows the visual efflorescence changes of RPSG with different alkali equivalents at 1 day, 3 days and 7 days. It can be seen from Figure 12 that after 1 day of immersion, the specimens RP-S30-M1.3N3 and RP-S30-M1.3N6 did not produce obvious crystalline substances, while the specimens RP-S30-M1.3N9, RP-S30-M1.3N12 and RP-S30-M1.3N15 have a little crystalline material on the surface. After 3 days of immersion, crystalline substances appeared on the surface and surrounding of all the specimens, and the most crystalline substances appeared in the specimens RP-S30-M1.3N3 and RP-S30-M1.3N15. After 7 days of immersion, visual efflorescence of the specimens was more obvious. The activity of RP and slag is weaker at a low alkali equivalent, and the specimen has smaller strength and larger capillary porosity (as shown in Figure 9), so the rate of efflorescence of specimen RP-S30-M1.3N3 is faster. However, the specimen RP-S30-M1.3N15 with a high alkali equivalent also had an obvious efflorescence phenomenon after immersion in water for 1 day, and the degree of efflorescence was high. The reason for this is that specimens with a high alkali equivalent are not stable and the crystallization pressure generated by the excess residual alkali in the pores causes cracks to form in the top and bottom of the specimen. This leads to rapid and severe efflorescence, which eventually leads to the rupture of the specimen. According to the mass loss results in Figure 12, it can be seen that the relationship between efflorescence and alkali equivalent is not an absolute positive correlation. At lower alkali equivalents, the degree of efflorescence decreases as the alkali equivalent increases. For the specimens with high alkali equivalents, the higher the alkali equivalent, the higher the degree of efflorescence. In this paper, the 6 wt.% alkali equivalent specimen has the least efflorescence. From the results of the analysis based on Figure 9, it can be seen that the capillary porosity of the specimens RP-S30-M1.3N6, RP-S30-M1.3N9 and RP-S30-M1.3N12 is basically the same, but the relationship between the degree of efflorescence is: RP-S30-M1.3N12 > RP-S30-M1.3N9 > RP-S30-M1.3N6. This is because the modified sodium silicate solution with high alkali equivalent has a higher concentration of alkali metal ions.

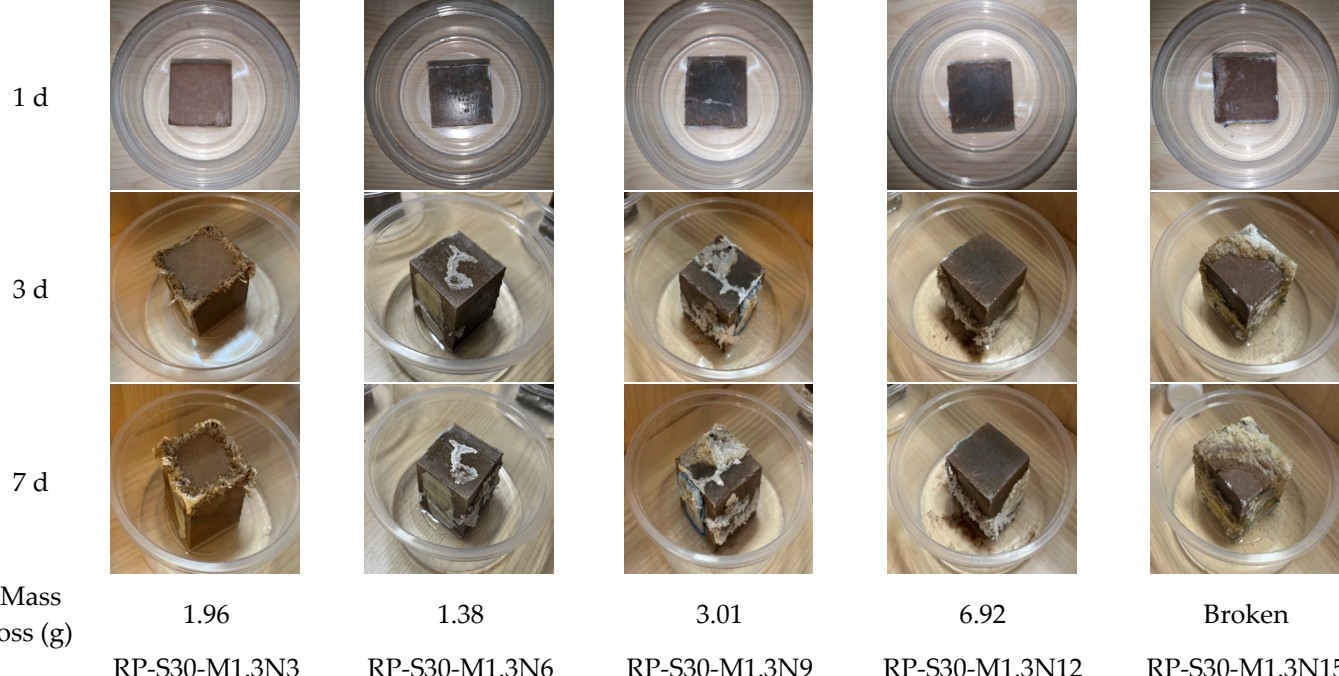

**Figure 12.** Visual efflorescence of RPSG with different alkali equivalents.

### 4.3.3. Effect of Modulus on Visual Efflorescence

Figure 13 shows the visual efflorescence changes of RPSG with different moduli at 1 day, 3 days and 7 days. It can be seen from Figure 13 that after immersion in water for

1 day, no obvious crystallization appeared in all the specimens. After immersion in water for 3 days, flower-like white crystals appeared about 5 mm above the immersed part of the bottom of all specimens, and this phenomenon was more obvious after immersion in water for 7 days. The mass losses of the specimens are all at small values, while the mass losses of the high modulus specimens RP-S30-M1.5N6 and RP-S30-M1.7N6 are significantly smaller than those of the low modulus specimens RP-S30-M0.9N6, RP-S30-M1.1N6 and RP-S30-M1.3N6. This is because the modified sodium silicate solution with high modulus contains more active $SiO_2$, which facilitates the formation of denser gels in the specimens, thereby improving the degree of efflorescence of the specimens [64]. This is well demonstrated by the analysis in Figure 5 above.

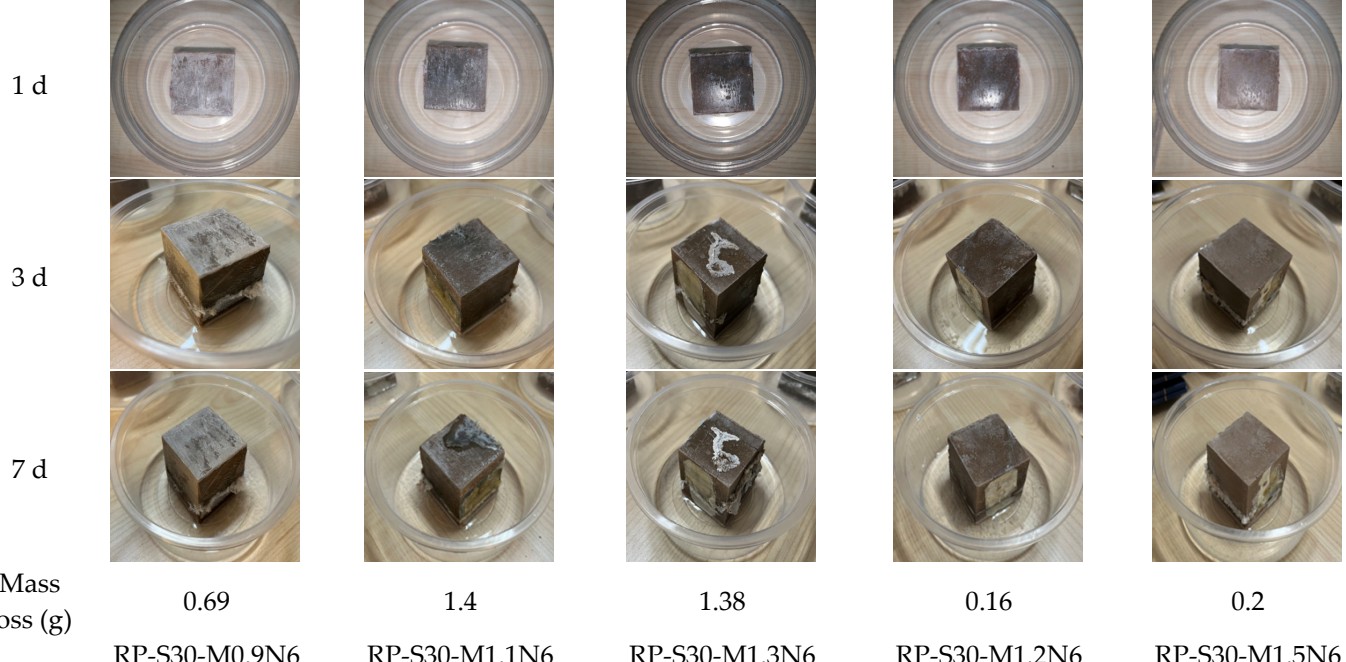

**Figure 13.** Visual efflorescence of RPSG with different moduli.

### *4.4. PH Value*

#### 4.4.1. Effect of Slag Content on pH Value

Figure 14 shows the pH values of RPSG with different slag contents. As shown in Figure 14, slag delayed the increase in pH value of the specimens. After 72 h of immersion, the pH values of the individual specimens gradually reach their maximum values with minimal differences. It suggests that most of the alkali metal ions in the specimens will be released from the specimens as the immersion time is extended. However, at 2 h of immersion, the pH values of the different specimens showed significant differences. The pH value of specimen RP-S0-M1.3N6 was the largest, and the pH value of specimen RP-S45-M1.3N6 was the smallest. It shows that the efflorescence rate of the specimens becomes slower with increasing slag content.

#### 4.4.2. Effect of Alkali Equivalent on pH Value

Figure 15 shows the pH values of RPSG with different alkali equivalents. As shown in Figure 15, the pH value of specimen RP-S30-M1.3N6 was the smallest (pH ≈ 9.4), and the pH value of specimen RP-S30-M1.3N15 was the largest (pH ≈ 11.4) at the 2 h immersion time. The results were validated against each other with the visual efflorescence results in Figure 12. After 24 h of immersion, all specimens reached a pH value above 10.0 and gradually increased in the later stages. Meanwhile, as can be observed in Figure 15, the pH value of the specimens increased with increasing alkali equivalents. It suggests that the

efflorescence potential of the specimens becomes higher with increasing alkali equivalents. However, this does not indicate the actual degree of efflorescence in relation to the alkali equivalents. According to the results in Figures 9 and 12, the efflorescence of the specimen is also influenced by the denseness of the matrix.

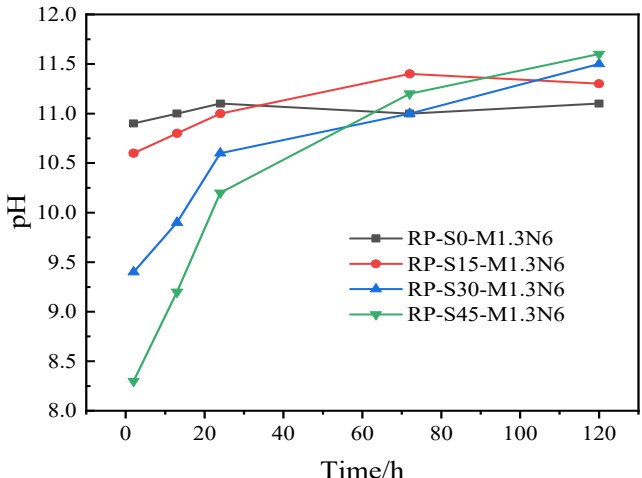

**Figure 14.** PH values of RPSG with different slag contents.

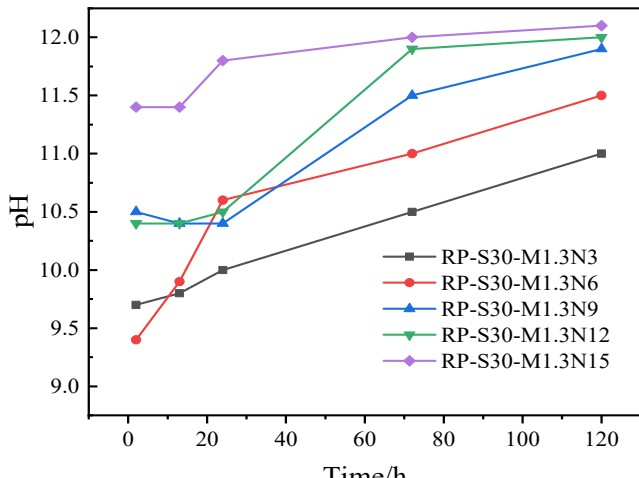

**Figure 15.** PH values of RPSG with different alkali equivalents.

### 4.4.3. Effect of Modulus on pH Value

Figure 16 shows the pH values of RPSG with different moduli. It can be observed from Figure 16 that the pH value of the lower modulus specimens is greater than that of the higher modulus specimens. It shows that increasing the modulus is an effective means of improving the efflorescence of the matrix. Meanwhile, it can be observed that the pH value of the specimens is slightly influenced by the modulus. It suggests that the modulus has a lower influence on the degree of efflorescence of RPSG.

### 4.5. Micro-Analysis

Based on the analysis results of drying shrinkage and efflorescence above, it is clear that the relationship of each factor on the durability performance of RPSG is: slag content > alkali equivalent > modulus. Meanwhile, the changes in alkali equivalent and modulus influence the durability performance of RPSG by inhibiting or promoting the hydration reaction of RP and slag. Therefore, the specimens with different slag contents were selected for further micro-analysis in this paper.

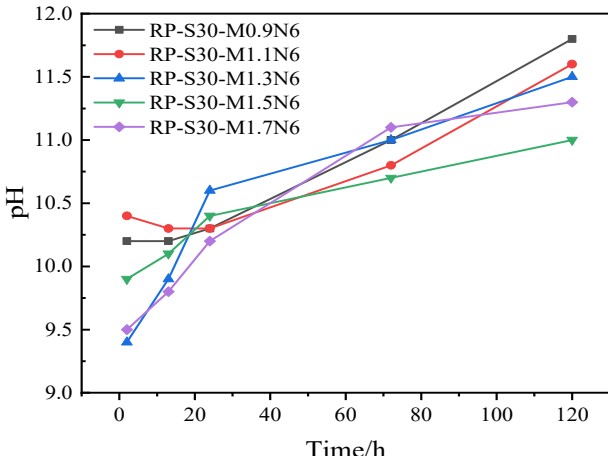

**Figure 16.** PH values of RPSG with different moduli.

### 4.5.1. XRD

Figure 17 shows the XRD diffraction patterns of RPSG. According to Figure 17, compared with the specimen RP, the diffraction peaks of chlorite, mica and feldspar in RP-S0-M1.3N6 and RP-S30-M1.3N6 are significantly lower. This shows that the alkaline environment provided by the modified sodium silicate can dissolve the layered silicoaluminate minerals of chlorite in the RP. The modified sodium silicate can provide reaction conditions for the RPSG. The specimen RP-S30-M1.3N6 showed some broader diffraction peaks between 20° and 40°, but the phenomenon was not significant. This is because the slag prompted the generation of poorly crystalline hydration products CASH and NASH from the RPSG. The amorphous nature of these hydration products, coupled with their overlap with the diffraction peaks of the more crystalline minerals in the XRD diffraction pattern, means that they do not have a distinct diffraction peak in Figure 17. In addition, specimen RP-S30-M1.3N6 has a weakened diffraction peak in the calcite phase compared to specimen RP-S0-M1.3N6, which has a significant effect on drying shrinkage of the calcite relative to the slurry [68]. It should be noted that the diffraction peak intensity of quartz decreases with the addition of activator or slag, and this is mostly caused by the dilution effect [69]. In fact, the amount of crystalline quartz dissolved in an alkaline environment is so small that it has a negligible effect on the variation of its diffraction intensity.

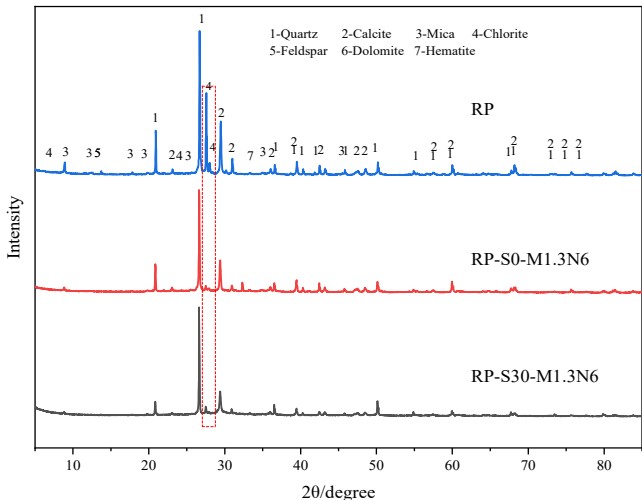

**Figure 17.** RD patterns of RPSG with different slag contents.

### 4.5.2. FTIR

Figure 18 shows the FTIR images of RPSG. As shown in Figure 18, significant variations can be observed around wave numbers 400 to 800 cm$^{-1}$ and 1000 cm$^{-1}$. For the spectrograms in the wave number range of 400 to 800 cm$^{-1}$, the wave peak of specimen RP-S30-M1.3N6 is significantly smaller than that of specimen RP-S0-M1.3N6. This wavenumber band corresponds to the vibrational band produced by SiO$_4$ and AlO$_4$ tetrahedra, suggesting that the addition of slag promotes the dissolution of the silicon aluminum component of the RP [20]. Around wave number 1000 cm$^{-1}$, the peak intensity of specimen RP-S30-M1.3N6 with the addition of slag was significantly weaker compared to that of specimen RP-S0-M1.3N6. At the same time, the wave peak of specimen RP-S30-M1.3N6 was shifted towards a lower wavenumber compared to that of specimen RP-S0-M1.3N6. The spectral band near 1000 cm$^{-1}$ corresponds to the stretching vibration of Si-O-Si and the asymmetric stretching vibration of T-O-Si (T is Si or O), indicating that the addition of slag promotes the dissolution of silica–aluminates and the hydration of low polymerization gel formation [70,71].

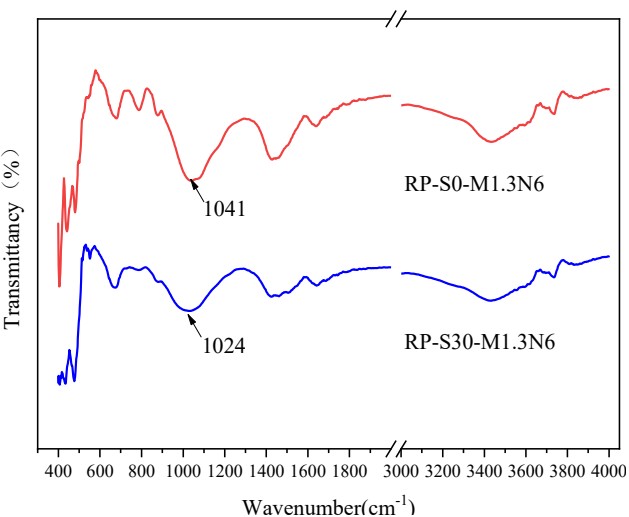

**Figure 18.** FTIR results of RPSG with different slag contents.

### 4.5.3. TG

Figure 19 shows the TG/DTG curves of RPSG. As can be seen from Figure 19a,b, the specimens have three main regions of mass loss corresponding to three temperature ranges: (1) room temperature ~ 200 °C, (2) 200–600 °C and (3) 600–1000 °C. Higher temperature results are not covered in this paper. The mass loss of the specimens below 200 °C is mainly the loss of free water adsorbed in the voids, while 200 to 600 °C is mainly the loss of bound water, corresponding to the dehydroxylation process of the hydration products NASH and CASH [72]. The mass loss of specimen RP-S30-M1.3N6 in this range is greater than that of specimen RP-S0-M1.3N6. This is because the incorporation of slag increased the internal hydrated gel content of the specimens, and more free water was adsorbed on the surface of the gel pores, while slag promoted the generation of CASH and NASH gels (corroborating the analysis in Figure 17). The mass loss of the specimen at 600–1000 °C is mainly the loss of calcium carbonate, which is closely related to the carbonation resistance of the cementitious material [15]. The mass loss rate of specimen RP-S30-M1.3N6 in this range is less than that of specimen RP-S0-M1.3N6, indicating that slag can improve the carbonation resistance of RPSG.

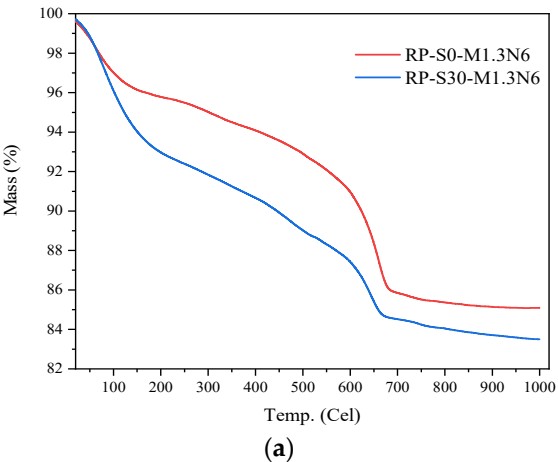
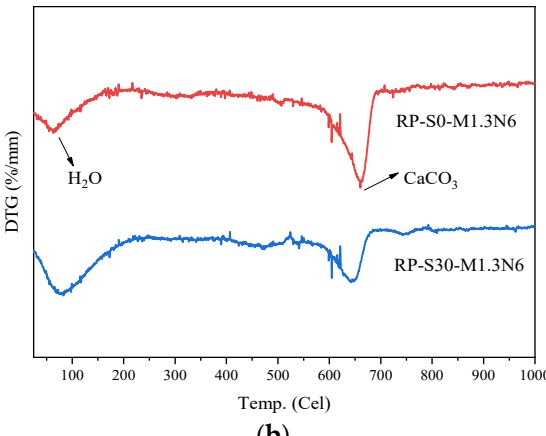

**(a)**  **(b)**

**Figure 19.** TG/DTG curves of RPSG with different slag contents.; (**a**) Mass loss curves of RPSG; (**b**) DTG curves of RPSG.

### 4.5.4. SEM

Figure 20 shows the SEM photos of RPSG with different slag contents after standard curing for 28 days under the condition of fixed alkali equivalent and modulus. It can be observed that the SEM photos of RP-S30-M1.3N6 show smaller pores, and the structure is denser. In combination with the analyses in Figures 17–19, it is clear that slag promotes the dissolution of the silica–aluminum component of the RPSG and the formation of hydrated gels such as NASH and CASH with low polymerization, resulting in a denser structure of the specimen RP-S30-M1.3N6. Some unbound loose stroma can be seen in the lower right corner of RP-S0-M1.3N6, while no similar result is observed in RP-S30-M1.3N6. This indicates that the slag particles in RP-S30-M1.3N6 and RP particles have been fully bonded after 28 days. This is because slag rapidly dissolves a large number of calcium ions in the alkaline environment, which rapidly reacts with active $SiO_2$ dissolved from the RP to form geopolymer gel with a network tetrahedron structure, making the matrix denser [73,74].

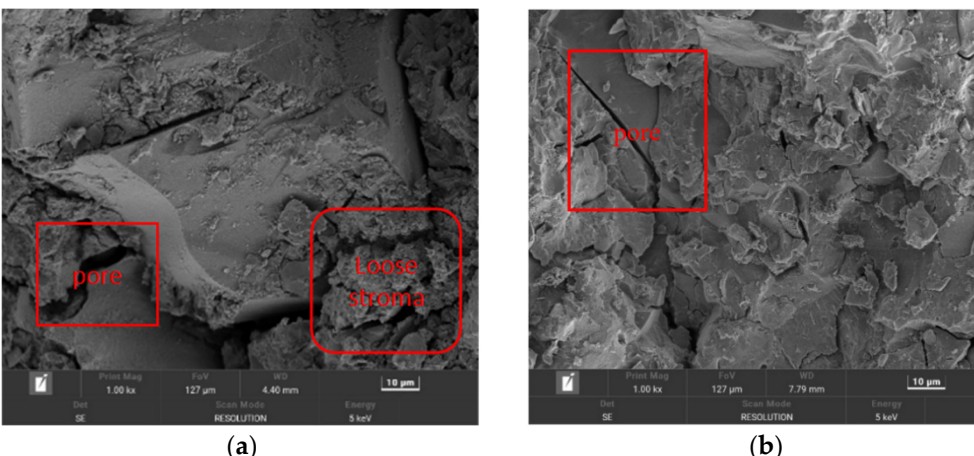

**(a)**  **(b)**

**Figure 20.** SEM images of RPSG with different slag contents [68,75]; (**a**) RP-S0- M1.3N6 (1000×); (**b**) RP-S30- M1.3N6 (1000×).

### *4.6. Discussion*

Currently, the international community does not have a uniform rule for measuring the degree of efflorescence of geopolymers [50,76,77]. This paper selected capillary porosity, visual efflorescence and PH value as indexes of efflorescence degree to ensure the conclusion is more reliable. Before the mix proportion design, the group conducted exploratory tests to ensure that the formed specimens had good mechanical and workability properties.

Meanwhile, the conclusions are valid within the limits of the paper. However, further study is needed to determine whether the conclusions continue to be valid if the limited range is exceeded, such as modulus higher than 1.7. In addition, the recycled micronized powder originated from construction waste disposal plants and mainly contained mortar, clay bricks, ceramics and tiles without further screening.

## 5. Conclusions

In this paper, the effects of slag content, alkali equivalent and modulus on drying shrinkage and efflorescence of RPSG under normal temperature curing were studied. The drying shrinkage pattern and efflorescence pattern of RPSG were systematically analyzed. The main conclusions of this paper are as follows:

(1) Slag significantly reduces the drying shrinkage and efflorescence of RPSG. Slag can promote RPSG to generate more hydration products, which causes the matrix to become denser and reduces the capillary porosity, thus significantly improving its drying shrinkage and efflorescence performance.

(2) When the modulus is constant, the potential for the efflorescence of RPSG increases with increasing alkali equivalent. However, the relationship between drying shrinkage and alkali equivalent is not a simple linear relationship. At low alkali equivalents (6% in this paper), the drying shrinkage of RPSG increases with increasing alkali equivalents and decreases with increasing alkali equivalents at high alkali equivalents.

(3) The drying shrinkage of RPSG increases with increasing modulus when the alkali equivalent is constant. In contrast, the degree of efflorescence decreases with increasing modulus.

(4) Micro-analysis has shown that slag promotes the dissolution of the silica–aluminum composition in RPSG and the formation of hydrated gels such as NASH and CASH with low polymerization. This results in a slurry with smaller pores and a denser structure.

The research in this paper shows that the effects of slag content, alkali equivalent and modulus on the drying shrinkage and efflorescence of RPSG are not identical. In this study, RP-S45-M1.3N6 is the best proportional design for RPSG. The drying shrinkage properties and efflorescence performance of RP-S45-M1.3N6 were significantly reduced compared with RP-S0-M1.3N6 (drying shrinkage was reduced by 76.32%, visual efflorescence was significantly alleviated, capillary porosity was reduced by 60.9% and early pH value was reduced by approximately 2.0). The research has demonstrated that geopolymers with excellent durability can be prepared by using slag and RP, which has a beneficial impact on promoting the resource utilization of RP.

The main materials used in the prepared RPSG are solid wastes, which have complex and variable chemical compositions. The effects of the content of the main chemical components, such as $CaO$, $Al_2O_3$ and $SiO_2$, on the properties of RPSG should be further investigated.

**Author Contributions:** Conceptualization, X.L.; methodology, X.L.; resources, X.L.; writing—original draft, X.L. and E.L.; supervision, X.L.; validation, E.L. and Y.F.; investigation, E.L. and Y.F.; data curation, E.L. and Y.F.; writing—review and editing, E.L. and Y.F. All authors have read and agreed to the published version of the manuscript.

**Funding:** This research received no external funding.

**Institutional Review Board Statement:** Not applicable.

**Informed Consent Statement:** Not applicable.

**Data Availability Statement:** The data presented in this study are available on request from the corresponding author.

**Acknowledgments:** This research did not receive any specific grant from funding agencies in the public, commercial, or not-for-profit sectors.

**Conflicts of Interest:** The authors declare no conflict of interest.

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
