# Peer review of "Reduction in Drying Shrinkage and Efflorescence of Recycled Brick and Concrete Fine Powder–Slag-Based Geopolymer"

_applsci, doi:10.3390/app13052997_

Round 1

Reviewer 1 Report

The paper is written in a clear manner and generally corrected form, but there are some language and grammar mistakes spread throughout the paper. Thus, the Reviewer thinks that the Authors should read carefully the paper to correct the above mistakes.

Page 1-Line 12

The word “and” is missing between “brick” and “concrete”.

Page 1-Lines 37-38

Maybe “stacking and landfill” should be replaced by “stacking in landfill”.

Page 1-Line 43

Recycled aggregates are produced not only from construction waste but also from demolition waste, which constitutes a not negligible part of total waste. Authors should highlight that some of demolition waste is derived from the demolition of buildings damaged by fire, as the one considered in the following research studies:

Frappa G, Pauletta M, Pitacco I, Russo G. Experimental tests for the assessment of residual strength of r.c. structures after fire – Case study. Engineering Structures 2022;252.

doi: 10.1016/j.engstruct.2021.113681

Wagih AM, El-Karmoty HZ, Ebid M, Okba SH. Recycled construction and demolition concrete waste as aggregate for structural concrete. HBRC Journal Volume 9, Issue 3, December 2013, Pages 193-200.

Rao A, Jha KN, Misra S. Use of aggregates from recycled construction and demolition waste in concrete. Resources, Conservation and Recycling Volume 50, Issue 1, March 2007, Pages 71-81

Page 1- Line 44

With regard to recycled and sustainable concretes, the following research works can be cited, due to their relevance in the topic:

Afroughsabet V, Biolzi L, Monteiro PJM, Gastaldi MM. Investigation of the mechanical and durability properties of sustainable high performance concrete based on calcium sulfoaluminate cement. Journal of Building Engineering 2021;43, article number 102656.

Biolzi L, Cattaneo S, Guerrini G, Afroughsabet V. Sustainable concretes for structural applications. Research for Development 2020, pp 249 – 261.

Page 2- Line 53

With regard to Portland cement the following research work is relevant to be mentioned:

Frappa G, Miceli M, Pauletta M. Destructive and non-destructive tests on columns and cube specimens made with the same concrete mix. Construction and Building Materials 349 (2022) 128807.

Page 2- Lines 64-65

It should be highlighted that drying shrinkage reduces the durability and applicability not only of concrete structures but also of all historical assets whose stability relies on mortar connections. The great importance of this issue is well outlined in the following study:

Frappa G, Pauletta M, Gaetano R. Failure analysis of three rigid block assemblies – A real collapse

resulting in death. Engineering Failure Analysis 145 (2023) 107001.

Page 2- Line 79

The following part of sentence is not correctly worded

It often used additives include magnesium oxide

Page 2-Line 98

were” should be corrected in “was”.

Page 3-Line 102

however” is redundant.

Page 3-Line 108

were less efflorescence” is not correctly worded.

Page 3- Lines 127-131

With regard to the problem of efflorescence in geoplymers, some bibliography should be added. For instance the following work can be considered:

De Oliveira LB, De Azevedo A RG, Marvila MT, Pereira EC, Fediuk R, Vieira CMF. Durability of geopolymers with industrial waste. Case Studies in Construction Materials, Volume 16, June 2022, e00839.

Page 3-Line 129

The meaning of acronyms XRD, FTIR, TG and SEM should be explicited.

Page 6- Lines 205-207

The following sentence is not correctly worded

Scraped  out  the  white  crystals  and  … weighed its mass”.

Page 8-Line 263

Figure 17 does not show the density change.

Figure 10 and Figure 13

In the abscissa axis of these figures, it is indicated that time is expressed in hours. Is it correct or the reported time values are expressed in minutes?

Page 13-Line 434

The name of specimen RP-S30-M1.3N13 should be corrected in RP-S30-M1.3N12.

Page 14-Lines 479-481

Check that the data reported in this sentence “the pH value of specimen RP-S30-M1.3N6 was the smallest (pH ≈ 9.0) and the pH value of specimen RP-S30-M1.3N6 was the largest (pH ≈ 11.4) at the 2 hours immersion time.”

Page 15-Line 494

Maybe Figure 10 is cited instead of Figure 14. Check it.

Page 19-Lines 615-616

The following sentence is not correctly worded

This is different from efflorescence, whose drying shrinkage is not simply positively correlated with the alkali equivalent”.

Author Response

Thank you very much for your valuable and helpful comments. Your suggestions are really valuable and helpful for revising and improving our paper. The authors need to explain that we have reformatted the original chapter 3.2 based on the comments of reviewer 2 and reviewer 4. In order to facilitate the reviewers' and editors' access, the "track changes" function was not used during the reformatting (some figure names were changed) and reference changes. Besides that, all changes have been marked using the "track mark" function. According to your suggestions, we have made the following revisions on this manuscript.

Reviewer 2 Report

In this paper, the effects of some parameters such as slag content and alkali equivalent under normal temperature curing were studied experimentally. The subject matter is interesting. The following comments may help the authors to improve their work:

1-     It would be better if the authors compare the results of their experimental tests with the existing codes/regulations/literature that estimate the considered parameters of this article (e.g., shrinkage). It may help us to have a good overview about the content of this article and what the others have published before.

1-     How slag could promote RPSG? Please mention the response within the article.

2-     Please add reference of Fig. 3. The same issue for Fig. 20.

3-     After introduction, please add a new section entitled “research significance”. In this new section, please explain the novelty and research significance in one or two paragraphs.

4-     If the modulus is not constant, what changes have been occurred in potential for the efflorescence of RPSG? If it increased, is this conclusion valid in whole ranges of modulus? Please mention the response within the article.

2-     Section 3-2 should be itemized.

3-     Both axes should have a clear title. For example, see Fig. 18. Please resolve this issue in the whole figures.

4-     Before conclusion, please add a discussion section. Some information such as limitations, pros and cons, could be mentioned and discussed in this new section.

Reviewer 3 Report

This paper investigates reduction in drying shrinkage and efflorescence of recycled brick and concrete fine powder-slag based geopolymer. The study is well written, and the discussion part is sufficient. The following corrections are required for the study to be published:

·       The introduction section can be shortened and especially the more important parts can be emphasized.

·       Line 143: “Table 1. Table 1 The chemical composition of RP and slag.” Revise the sentence.

·       In Figure 1, “Passing 2.36 mm square hole sieve” is written both in the first and last stages of the production phase. The sieve size in the last stage is misspelt and should be corrected.

·       Were the test results presented in Fig.8, Fig.11 and Fig.14 performed on a single sample? If performed on more than one sample, standard error bars should be added to the graphs. I also recommend removing the data labels on these charts. The values are both presented in the text below and can be read from the "y-axis".

·       Instead of “Mass (g)” given at the bottom of Fig.9, Fig.12 and Fig.15, “Mass Loss (g)” should be written.

·       In the Conclusion section, suggestions should be given for future studies on this subject.

Reviewer 4 Report

In all, the manuscript is of high quality and will be a great addition to the body of knowledge on the subject of construction waste.

The manuscript requires the input of a senior faculty member or a professional English language editor to correct the syntax and grammatical errors.

Sentence one in Line 33-34 needs to be rewritten due to the poor sentence construct. The authors can use applications such as Grammarly to first address the obvious language issues associated with the manuscript.

The use of acronyms without a prior definition is wrong. The authors should address this especially for the acronyms in Line 129.

To avoid confusing the readers and for consistency, the numbering (1-9) under Section 2.3 should be changed to 2.3.1 - 2.3.9.

Are there any limitation or delimitation to the conduct of the research/experiment? Could the climatic condition of the region or research location contributed to the outcome or visual efflorescence of RPSG samples. The authors should provide brief statement around these questions  within the conclusion section and also provide recommendations which can be useful to other researchers who might want to repeat the same research in other geographical locations.

Round 2

Reviewer 1 Report

The Reviewer thinks that the paper is good for publication. 

Reviewer 2 Report

Accept as is.